# Frontoparietal connectivity as a product of convergent evolution in rodents and primates: functional connectivity topologies in grey squirrels, rats, and marmosets

David J. Schaeffer [1✉], Kyle M. Gilbert [2], Miranda Bellyou [2], Afonso C. Silva [1] & Stefan Everling [2,3]

Robust frontoparietal connectivity is a defining feature of primate cortical organization. Whether mammals outside the primate order, such as rodents, possess similar frontoparietal functional connectivity organization is a controversial topic. Previous work has primarily focused on comparing mice and rats to primates. However, as these rodents are nocturnal and terrestrial, they rely much less on visual input than primates. Here, we investigated the functional cortical organization of grey squirrels which are diurnal and arboreal, thereby better resembling primate ecology. We used ultra-high field resting-state fMRI data to compute and compare the functional connectivity patterns of frontal regions in grey squirrels (*Sciurus carolinensis*), rats (*Rattus norvegicus*), and marmosets (*Callithrix jacchus*). We utilized a fingerprinting analysis to compare interareal patterns of functional connectivity from seeds across frontal cortex in all three species. The results show that grey squirrels, but not rats, possess a frontoparietal connectivity organization that resembles the connectivity pattern of marmoset lateral prefrontal cortical areas. Since grey squirrels and marmosets have acquired an arboreal way of life but show no common arboreal ancestor, the expansion of the visual system and the formation of a frontoparietal connectivity architecture might reflect convergent evolution driven by similar ecological niches in primates and tree squirrels.

[1] Department of Neurobiology, University of Pittsburgh, Pittsburgh, PA, USA. [2] Centre for Functional and Metabolic Mapping, Robarts Research Institute, University of Western Ontario, London, ON, Canada. [3] Department of Physiology and Pharmacology, University of Western Ontario, London, ON, Canada. ✉email: dschaeff@pitt.edu

The prefrontal cortex (PFC) is a granular neocortical region involved in complex cognitive functions[1,2]. Reciprocal connectivity between PFC and the posterior parietal cortex (PPC) is a defining feature of primate cortical organization[3–6]. Whether mammals outside of the primate order possess similar organization of frontoparietal connectivity is a controversial and still unresolved topic. The closest order to primates is Scandentia, made entirely of tree shrews. Tree shrews have a small granular PFC and form frontoparietal connectivity, but parietal visual areas connect with secondary motor areas and not the granular PFC[7]. Together with flying lemurs (order Dermoptera) and primates, Scandentia are members of the grand order Euarchonta, which separated from the ancestor of rodents and lagomorphs (together forming the grand order Glires) about 80 million years ago[8] (see Fig. 1 for phylogenetic tree). Here, we set out to better understand how frontoparietal functional connectivity in a mammal outside of the primate order—the Eastern Gray Squirrel (grand order Glires)—compares to both primates (grand order Euarchonta) and rodents (grand order Glires). Specifically, by comparing patterns of connectivity in an animal that has evolved in a similar environmental niche to primates, we sought to gain insight on how their brain adapted within the constraints of its evolutionary history[9,10].

Despite their larger evolutionary distance from primates, rats (*Rattus norvegicus*) and mice (*Mus musculus*) are—for practical reasons—the most widely used experimental mammalian species for neuroscientific inquiry. These two rodent species, however, do not possess a granular PFC[4,11], and although the so-called medial PFC in these two species is often regarded as a functional homolog to the primate PFC[12], the anterior medial wall of the rodent PFC may better correspond to cingulate cortex than to lateral PFC in primates[12]. Most importantly, medial PFC areas in rats do not form strong functional connectivity with posterior parietal areas[11]. Such disparities in cortical organization have been interpreted as fundamental differences between primate and rodent brains[4].

Concomitant to a dissimilar cortical organization, rats and mice have fundamentally different behaviors and lifestyle from primates. Rats and mice are nocturnal, and their exploratory and navigation behavior (whisking), diets, locomotion, and environmental niche are profoundly distinctive from those of primates. However, rats and mice represent only a small sample of the entire order of rodents that comprises five suborders with 34 families, making up 40% of all mammalian species[13]. Therefore, it is prudent to investigate the functional cortical organization in a rodent species that has an ecological niche more like that of many primate species (See Table 1 for comparison).

The North American grey squirrel (*Sciurus carolinensis*) resembles the small New World marmoset monkey (*Callithrix jacchus*) in several major ways: they have a similar arboreal lifestyle, they both have long tails that help them keep their balance, their digits have sharp claws, and they have similar body sizes[14]. Like most primates, grey squirrels are diurnal. They have a much better developed visual system than nocturnal rats[15], including two-cone color vision[16,17], larger visual cortical areas 17 and 18[18], a 5-layered geniculate nucleus[19], and a superior colliculus that is about ten times bigger than would be expected for a rat of similar size[14,15]. Studies using histochemical tracing and electrical microstimulation in squirrels suggest that connectivity of somatosensory and motor cortex have an organization that may be more similar to primates than that of rats or mice[20]. Of particular interest are the relatively large PPC in squirrels and the cortex in front of the primary motor cortex, labeled area F[21]. This frontopolar cortical region has a clearly defined layer 4[21], potentially resembling the granular PFC in primates. Whether this region in squirrels forms connectivity with posterior parietal regions similar to that of the primate frontoparietal network is yet to be established. To address this question, we leveraged our recent developments in ultra-high field MRI hardware for small animals[22] to acquire high-quality resting-state (RS-fMRI) data in grey squirrels under light anesthesia at 9.4 Tesla and compared it to RS-fMRI data from rats and marmosets. RS-fMRI allowed us to compute the functional connectivity (FC) between brain areas, and the application of the fingerprinting technique[23,24] provided a method of comparing the similarity of interareal patterns of FC across the three species.

## Results

Whole-brain functional connectivity was calculated using species-specific frontal seed regions for grey squirrels, rats, and marmosets. The resultant group functional maps are shown in Supplementary Fig. 1 (rats), Supplementary Fig. 2 (squirrels), and Supplementary Fig. 3 (marmosets). For comparisons of interareal FC patterns between the three species, we computed the FC of these frontal seed regions with five cortical areas: primary motor (M1), insular cortex (Ins), primary somatosensory (S1), posterior cingulate cortex (PCC), and posterior parietal cortex (PPC), and three subcortical regions: striatum (Str), pulvinar (Pul), and superior colliculus (SC). Figure 2 shows the location of these eight regions and the corresponding normalized FC fingerprints for rats (Fig. 2a), grey squirrels (Fig. 2b), and marmosets (Fig. 2c).

To compare the FC fingerprints between the three species, we computed cosine similarities for pairs of fingerprints (Supplementary Fig. 4 (rats vs. marmosets), Supplementary Fig. 5 (squirrels vs. marmosets), and Supplementary Fig. 6 (rats vs. squirrels). As shown in Fig. 3a, the fingerprints of rats and marmosets were different, as indicated by the cosine similarity metrics that were below 0.75 for most fingerprint comparisons (33/40). The only fingerprint comparisons that yielded substantially larger cosine similarities (>0.8) were for marmoset ventral premotor area 6 VA with frontal rat brain regions. Most of the comparisons of the fingerprints were significantly different (25/40; $p < 0.05$; white stars). This finding demonstrates that frontal rat regions had different FC fingerprints from those of the marmoset lateral prefrontal cortex.

The comparison of FC fingerprints between frontal regions in grey squirrels and marmoset lateral frontal cortex (Fig. 3b) revealed several region pairs (17/48) with cosine similarity metrics higher than 0.75, indicating similar FC profiles between

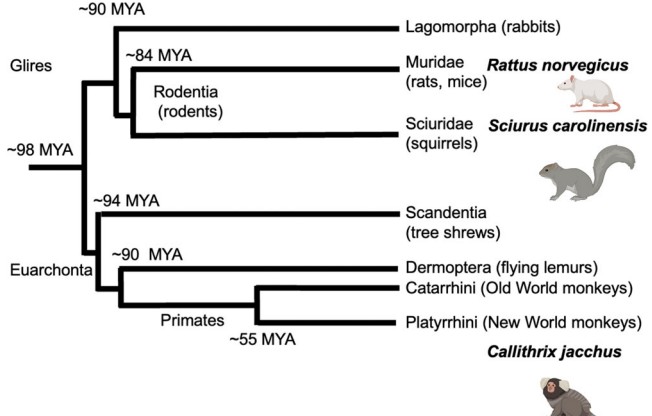

**Fig. 1 Phylogenetic tree showing the timelines of divergence (MYA: million years ago) and lineage of rats (*Rattus norvegicus*), squirrels (*Sciurus carolinensis*) and marmosets (*Callithrix jacchus*).** Based on ref. 49.

| | Rat (*Rattus norvegicus*) | Grey squirrel (*Sciurus carolinensis*) | Common marmoset (*Callithrix jacchus*) |
|---|---|---|---|
| **Table 1 Comparison of relevant features in each model species (similarities in bold).** | | | |
| Order | **Rodentia** | **Rodentia** | Primates |
| Sleep/wake cycle | Nocturnal | **Diurnal** | **Diurnal** |
| Navigation behavior | Whisking and olfaction | **Vision** | **Vision** |
| Environment | Terrestrial | **Arboreal** | **Arboreal** |
| Body mass | **250–500 g** | **300–710 g** | **250–650 g** |
| Brain mass | 1.8 g | **7.6 g** | **7.8 g** |

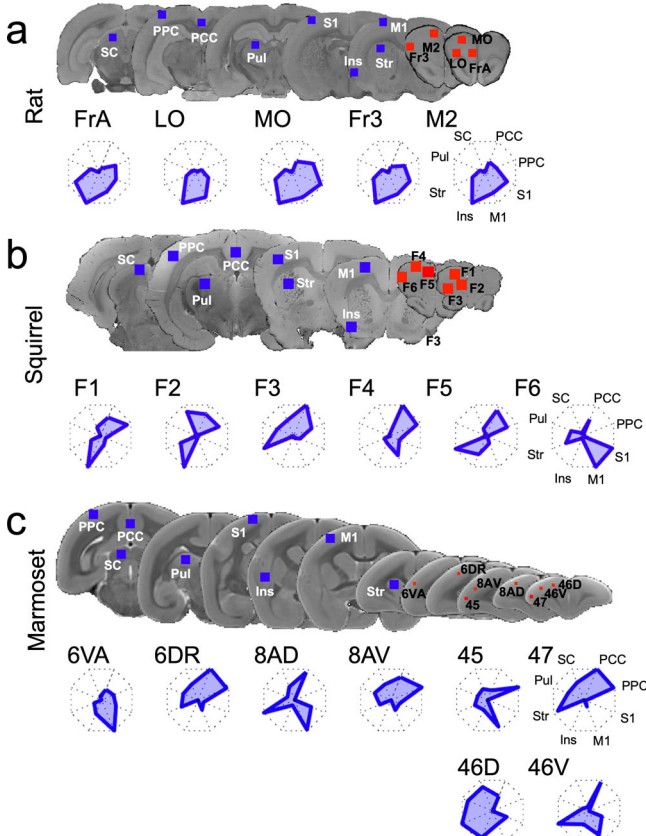

**Fig. 2 Region of interest (ROI) locations and fingerprint plots. a** ROI locations for rats. *Top*, seed ROIs (all in the right hemisphere) are shown in red, and 8 ROIs for the fingerprint analysis are shown in blue and overlaid on coronal slices of anatomical MR images. *Bottom*, fingerprint spider plots for frontal seeds with the posterior cingulate cortex (PCC), posterior parietal cortex (PPC), the primary somatosensory cortex (S1), the primary motor cortex (M1), insular cortex (Ins), striatum (Str), pulvinar (Pul), and superior colliculus (SC). **b** Same as (**a**), but for squirrels. **c** Same as (**a**), but for marmosets. Target region abbreviations: Frontal association cortex (FrA); lateral orbital cortex (LO); medial orbital cortex (MO); frontal cortex, area 3 (Fr3); secondary motor cortex (M2); frontal areas 1–6 (F1–F6); area 6 of cortex, ventral, part a (6 VA); area 6 of cortex, dorsorostral part (6DR); area 8a or cortex, dorsal part (8AD); area 8a of cortex, ventral part (8AV); area 45 of cortex;[46] area 47 of cortex;[48] area 46 of cortex, dorsal part (46D); area 46 of cortex, ventral part (46 V).

frontal regions in squirrels and lateral frontal areas in marmosets. Many comparisons were still significantly different between the two species (29/48; $p < 0.05$). The largest similarities were found between squirrel areas F3–F6 with lateral prefrontal marmoset areas. This finding shows that the squirrel frontoparietal cortex FC fingerprints resemble those of lateral prefrontal cortical areas in the marmoset.

Comparisons of the FC fingerprints of frontal regions in squirrels and rats confirmed overall low cosine similarities below 0.75 in 24/30 region pairs. The squirrel region F1 presented the largest similarities with the rat frontal regions ($p > 0.05$), while region F3 presented the largest differences with rat frontal regions FrA, LO, MO, and M2 ($p < 0.05$).

Figure 4a shows the FC fingerprint comparison of squirrel area F3 and marmoset area 47 L. Both regions show strong FC with the PCC and PPC at the cortical level and FC with striatum (Str), pulvinar (Pul), and superior colliculus (SC) at the subcortical level. These frontal regions are located on the anterior lateral surface (Fig. 4b). Figure 4c shows the FC map of squirrel region F3 with the rest of the brain. FC was strong with parietal medial area Pm and area L (limbic). FC of region F3 also extended into temporal areas Tm and Tp. In addition, there was significant FC with the striatum, pulvinar, and superior colliculus. The FC map of marmoset area 47 L (Fig. 4d) showed similar findings, with strong FC of area 47 L with PCC and PPC (mainly area LIP) as well as FC with the temporal cortex (area TE2). The map also shows some FC with the SC, Pul, and Str (caudate nucleus).

## Discussion

A frontoparietal network in which parietal visual areas have extensive reciprocal connections with the lateral prefrontal cortex is a defining feature of primate cortical organization[4,6]. Whether mammals outside the primate order possess a similar well-developed frontoparietal connectivity remains unanswered. We compared the functional connectivity (FC) patterns of frontal regions in grey squirrels, rats, and marmosets using RS-fMRI data to address this question. Our results demonstrate that grey squirrels have frontoparietal connectivity topologies that better overlap with that of primates than to that of rats.

Previously, a default-mode network consisting of a parietal subsystem clustered around the retrosplenial cortex and a temporal-frontal subsystem that included the auditory and orbitofrontal cortex has been described in rats[22]. However, the connectivity strength between these two subsystems was weak. Our rat data, which also show FC of area LO and MO with the retrosplenial cortex (Supplementary Fig. 1), is consistent with that previous report[22]. However, the FC of these frontal areas was much stronger with the insular cortex, M1, and S1 (Fig. 2a). These FC patterns are in stark contrast to the FC fingerprints of lateral prefrontal areas in marmoset (Fig. 2c), which showed strong FC with parietal areas.

A commonly held assumption is that the medial frontal cortex (mPFC) in rodents is homologous to the dorsolateral PFC in primates[12]. Our recent analysis of interareal fingerprints from RS-fMRI data in anesthetized marmosets, anesthetized rats, and awake humans did not support this idea[25]. Interareal fingerprints of medial frontal areas in rats resembled those of medial frontal areas, i.e., cingulate cortical areas in marmosets, but not dorsolateral PFC areas. Consistent with our previous report, we also found here that the connectivity pattern of frontal rat areas resembles those of area 6 VA in the ventral premotor cortex in

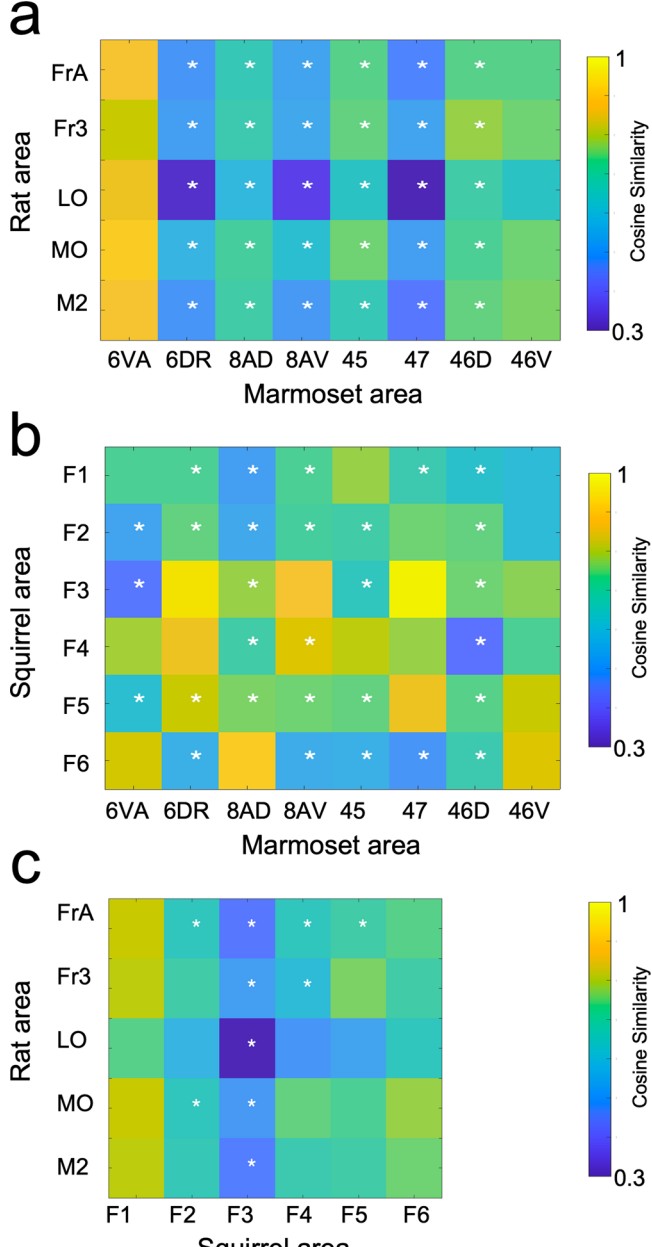

**Fig. 3 Similarity of interareal functional connectivity patterns between the species. a** Comparison of interareal functional connectivity patterns between rats and marmosets. For each species, cosine similarity values are plotted in matrix form. A high cosine similarity value suggests that the connectivity values are more comparable than a lower cosine similarity value. Significant differences are marked by a white asterisk within the similarity matrix. **b** Same as (**a**), but for comparing squirrels with marmosets. **c** Same as (**a**), but for comparing rats with squirrels. Seed region abbreviations: Frontal association cortex (FrA); lateral orbital cortex (LO); medial orbital cortex (MO); frontal cortex, area 3 (Fr3); secondary motor cortex (M2); frontal areas 1–6 (F1–F6); area 6 of cortex, ventral, part a (6 VA); area 6 of cortex, dorsorostral part (6DR); area 8a or cortex, dorsal part (8AD); area 8a of cortex, ventral part (8AV); area 45 of cortex; area 47 of cortex; area 46 of cortex, dorsal part (46D); area 46 of cortex, ventral part (46 V).

awake marmoset. This finding supports the notion that frontal areas in rats resemble primate premotor but not the prefrontal cortex (Fig. 3a; also see ref. 25).

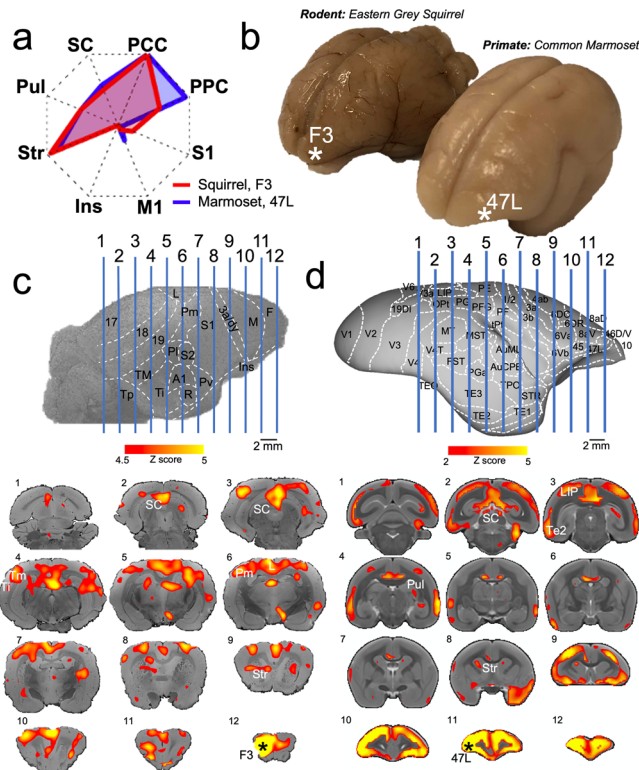

**Fig. 4 Similarities in interareal functional connectivity of squirrel region F3 and marmoset area 47L. a** Fingerprint for squirrel region F3 and marmoset area 47 L shows a similar interareal functional connectivity pattern. Note that for each species, the fingerprints are normalized between 0 and 1 to allow for pattern comparability: A 0 value does not necessarily mean that there was no activation in a region, but just that the region had the lowest relative value within that fingerprint. **b** Photograph of ex vivo squirrel and marmoset brains and the location of squirrel region F3 and marmoset area 47 L. **c** Anatomically annotated surface rendering and functional connectivity map of squirrel region F3 with the rest of the brain and overlaid on coronal slices of anatomical MR images. **d** Anatomically annotated surface rendering and functional connectivity map of marmoset area 47 L with the rest of the brain overlaid on coronal MR atlas slices.

The pattern, however, was entirely different for the comparison of connectivity profiles between squirrels and marmosets. We found that frontopolar regions in the squirrel showed high cosine similarities values that, unlike rats, did not significantly differ from patters of lateral frontal cortex areas 6DR, 8aD, 8aV, and 47 in marmosets. A prominent interareal connectivity feature of these PFC areas is a strong FC profile with posterior parietal cortex and posterior cingulate cortex. This connectivity profile was also present for frontopolar cortical sites in the squirrels. In contrast to rats and mice, the frontal cortex in squirrels has so far only been coarsely parcellated. The original report labeled the squirrel frontal cortex as a single area F[16]. More recently, Wong and Kaas (2008) divided the frontal cortex in squirrels into an agranular primary motor field (area M) but left the remaining cortex rostral and medial to area M as area F[26]. They noted that this region was not sharply distinguished from area M based on immunohistochemical and histochemical procedures. Interestingly, the frontopolar cortex in squirrels clearly shows a layer 4 that is lacking in rats and mice. Based on this cytoarchitectural feature and the strong FC with posterior parietal and posterior cingulate cortex, we propose that these frontal regions in squirrels could be considered analogous to some PFC regions in primates.

Eastern grey squirrels rely greatly on vision for climbing and jumping between tree branches and identifying potential predators. It is therefore not surprising that squirrels have a well-developed visual system. Hall and Diamond (1969) previously concluded that the visual system in grey squirrels exhibits a similar expansion as in tree shrews[27], an order closely related to primates. Squirrels have a retina with a 2:3 ratio of rods to cones[16], two types of cones with peak sensitivities in the green and blue color range[28], a lateral geniculate nucleus with five distinct layers[29], a larger pulvinar[30–32], a proportionally very large superior colliculus (SC)[17], and extensive visual cortical areas 17 and 18[18]. In contrast to the SC in rats, the squirrel SC receives abundant inputs from visual areas and cingulate cortex but not from the motor and somatosensory cortex, resembling the corticotectal pattern found in tree shrews and primates[33]. In fact, it has been proposed that diurnal squirrels may be the ideal rodent model for examining the visual cortex[15].

The frontoparietal connectivity that we identified here could provide the neural basis for the cognitive abilities found in squirrels. Under experimental conditions, squirrels exhibit key components of complex cognition, including behavioral flexibility in discrimination-reversal tasks[34,35], spatial cue use[36], and problem-solving strategies[37]. A fascinating natural example of inhibitory control in grey squirrels is that they stop digging and increase the latency to start caching when conspecifics are present[38].

Grey squirrels and marmosets have arrived at similar semi-arboreal lifestyles. However, they show no common arboreal ancestor, indicating that the expansion of the visual system and the formation of frontoparietal connectivity architecture might have evolved along similar lines driven by comparable ecological niches as an example of convergent evolution[27]. A diurnal activity pattern and an arboreal habitat may have favored color vision, high visual acuity, and depth perception for navigation in both species. At present, it cannot be ruled out that a frontoparietal network organization is typical in many rodent species and that the subfamily Muridae, which includes Old World rats and mice, lost this organization due to their adaptation to a terrestrial and mainly nocturnal lifestyle. It has been recently posited that evolution of the dorsal and ventral visual processing streams in both primates and squirrels was the result of occupying new, diurnal environments after the extinction of dinosaurs[14].

Rats and mice are powerful animal models for neuroscience. They are easy to breed, and powerful genetic tools have been developed that allow targeted probing and dissection of circuit functions in mice. Our findings, however, do not support the idea that rats have an organization of frontoparietal connectivity similar to primates. Based on the results presented here—that strong frontoparietal connectivity is present in *Sciurus carolinensis*, but is lacking in their Muridae counterparts, *Rattus norvegicus* (at least to the extent that it overlaps with primate connectivity patterns)—we highlight the importance of environmental niche in neuroscientific modelling species despite the constraints of evolutionary history. Indeed, evolutionary proximity is critically important, but evolutionary convergence may also be leveraged, perhaps improving the potential translatability of results yielded from preclinical modelling species. This rings especially true for inquiries in visual neuroscience. Indeed, even though squirrels and primates have been evolving independently for nearly 100 million years (Fig. 1) from ancestral insectivores[39,40] we found that grey squirrels show a frontoparietal architecture with comparable patterns to the marmoset brain, opening the possibility of using squirrels as a rodent model for vision and comparative explorations of frontoparietal network organization and function.

## Methods

### Subjects

*Grey squirrels and rats.* Data were collected from five (1 female, 4 melanistic) adult wild-caught Eastern grey squirrels (*Sciurus carolinensis*) of unknown age weighing 590–700 g under a wildlife scientific collector's authorization from the Ontario Ministry of Natural Resources. Once a squirrel was caught in a commercial trap (Havahart Squirrel & Chipmunk Live Trap), the trap was placed in a large plastic storage box and immediately transported to the Centre for Functional and Metabolic Mapping at the University of Western Ontario (5-min transport time). Squirrel anesthesia was induced by administering 4–5% isoflurane and oxygen with a flow rate between 1 and 1.5 l/min using a precision vaporizer attached to the plastic storage box, which served as an induction chamber. Data were collected from five adult male Wistar rats aged 8–12 weeks and weighing from 250–350 g. Rat anesthesia was induced by placing the animals in an induction chamber with 4–5% isoflurane and oxygen with a flow rate between 1 and 1.5 l/min.

Both species were lightly anesthetized via spontaneous inhalation of 1.5–2% isoflurane mixed with oxygen flowing between 1.5 and 2 L/min throughout the scan. Respiration, SpO2, and heart rate were continuously monitored using a pulse oximeter and were observed to be within the normal range throughout the scans (squirrels: respiration = 35–80 (mean 57) bpm, SpO2 = 95–100%, heart rate = 230–318 (mean 274) bpm). Body temperature was also measured and recorded throughout, maintained using warm-water circulating blankets, thermal insulation, and warmed air. All animals were head-fixed in stereotactic position using a custom-built MRI bed with ear bars and a species-specific palate bar as part of the anesthesia mask[22]. Upon completion of imaging, 3–5 ml of warmed NaCl was administered subcutaneously. A small ear punch was made in squirrels to mark the animal and avoid repeated study of the same squirrel. A blanket and supplemental heat were provided for recovery. After recovery, squirrels were transported in the plastic storage/induction box back to the capture location and released. One squirrel died during recovery. Its brain was removed, fixed by submersion in formalin, and subsequently imaged for ultra-high resolution structural MRI at the University of Pittsburgh Brain Institute. All experimental procedures were in accordance with the Canadian Council of Animal Care policy and protocols approved by the Animal Care Committee of the University of Western Ontario Council on Animal Care.

*Marmosets.* To compare the squirrel and rat data to marmoset data, we used fMRI data from our open-access resource (https://marmosetbrainconnectome.org)[41]. This database contains resting-state fMRI data from 31 awake marmosets (*Callithrix jacchus*, 8 females; age: 14–115 months; weight: 240–625 g) that were acquired at the University of Western Ontario (5 animals) and the National Institutes of Health (26 animals).

### Imaging

*Image acquisition.* For both rats and grey squirrels, data were acquired on a 9.4 T 31 cm horizontal bore MRI scanner (Varian/Agilent, Yarnton, UK) and Bruker BioSpec Avance III console with the software package Paravision-6 (Bruker BioSpin Corp, Billerica, MA) and a custom-built, high-performance 15-cm-diameter gradient coil with 400-mT/m maximum gradient strength[42] at the Centre for Functional and Metabolic Mapping at the University of Western Ontario. The animal holders and radiofrequency receive arrays were built in-house, and design files for these stereotactic holders have been made available[22] with geometrically optimized phased array receive coil designs for the two species. The rat coil was made up of 6 channels, while a larger marmoset coil with 8 channels was used for the squirrels. Both coils have a similar signal-to-noise ratio (SNR). Preamplifiers were located behind the animals, and the receive coil was placed inside a quadrature birdcage coil (12-cm inner diameter) used for transmission.

For the rats, functional imaging was acquired during one session for each animal, with 6 functional runs (at 600 volumes each) with the following parameters: TR = 1,500 ms, TE = 15 ms, field of view = 38.4 × 38.4 mm, matrix size = 96 × 96, voxel size = 0.4 × 0.4 × 0.4 mm, slices = 35, bandwidth = 280 kHz, GRAPPA acceleration factor: 2 (anterior-posterior). T2-weighted structural scans were acquired for each animal with the following parameters: TR = 7,000 ms, TE = 44 ms, field of view = 38 × 38 mm, matrix size = 192 × 192, voxel size = 0.2 × 0.2 × 0.4 mm, slices = 35.

The squirrel imaging was similar to rat imaging with functional imaging data acquired from one session for each animal with 3–4 functional runs (at 600 volumes each) with the following parameters: TR = 1500 ms, TE = 15 ms, flip angle = 40 degrees, field of view = 64 × 64 mm, matrix size = 128 × 128, voxel size = 0.5 × 0.5 × 0.5 mm, slices = 42, bandwidth = 400 kHz, GRAPPA acceleration factor: 2 (anterior-posterior). T2-weighted structural scans were acquired for each animal with the following parameters: TR = 7000 ms, TE = 55 ms, field of view = 64 × 64 mm, matrix size = 384 × 384, voxel size = 0.133 × 0.133 × 0.5 mm, slices = 45. As an in vivo registration template, we also acquired a structural T2-weighted image from squirrel #5 with an isotropic voxel resolution with the following parameters: TR = 15,000 ms, TE = 48 ms, field of view = 51 × 51 mm, matrix size = 256 × 256, voxel size = 0.2 × 0.2 × 0.2 mm, bandwidth = 50 kHz, slices = 100, number of averages = 3.

For the squirrels, an ultra-high-resolution ex vivo template was also generated from squirrel #5 – the same brain used as the in vivo registration template. The

sample was fixed by submersion in 10% formalin for one week, then imaged at the University of Pittsburgh Brain Institute on a 9.4 T 30 cm horizontal bore MRI scanner (Bruker BioSpin Corp, Billerica, MA) equipped with a Bruker BioSpec Avance Neo console and the software package Paravision-360 (version 3.2; Bruker BioSpin Corp, Billerica, MA) and a custom high performance 17 cm gradient coil (Resonance Research Inc, Billerica, MA) performing at 450 mT/m gradient strength. To maximize sensitivity, a custom 30 mm inner diameter millipede quadrature coil (ExendMR LLC, Milpitas, CA) was used. The sample was placed in a 50 ml conical centrifuge tube, filled with Fomblin oil (Solvay Solexis, West Deptford, NJ), and placed under vacuum (−27 inHg) for 30 min to remove air bubbles. A T2*-weighted structural scan was acquired with the following parameters: TR = 100 ms, TE = 8 ms, field of view = 35 × 30 × 20 mm, matrix size = 700 × 600 × 400, voxel size = 0.05 × 0.05 × 0.05 mm, bandwidth = 50 kHz, number of averages = 2, total scan time = 14 h, 48 min. The T1 relaxation time of the tissue was measured (mean T1 = 1047 ms), and an optimum flip angle (Ernst angle) of 24.7 degrees was set for a TR of 100 ms.

For the marmosets, functional imaging was acquired from two sites. The University of Western Ontario data were acquired using the same 9.4 T scanner as the rat and squirrel data. A custom 5-channel receive coil was used, which was rigidly fixed to the head implant[43]. Radiofrequency transmission was accomplished with a quadrature birdcage coil (12-cm inner diameter) built in-house. Functional imaging was performed over multiple sessions (days) for each animal, with 4-6 functional runs (at 600 volumes each) per animal with the following parameters: TR = 1500 ms, TE = 15 ms, flip angle = 35 degrees, field of view = 64 × 64 mm, matrix size = 128 × 128, voxel size = 0.5 × 0.5 × 0.5 mm, slices = 42, bandwidth = 500 kHz, GRAPPA acceleration factor: 2 (anterior-posterior). T2-weighted structural scans were acquired for each animal during one of the awake sessions with the following parameters: TR = 5,500 ms, TE = 53 ms, field of view = 51.2 × 51.2 mm, matrix size = 384 × 384, voxel size = 0.133 × 0.133 × 0.5 mm, slices = 42, bandwidth = 50 kHz, GRAPPA acceleration factor: 2.

The NIH marmoset data were acquired using at 7 T 30 cm horizontal bore magnet (Bruker BioSpin Corp, Billerica, MA, USA) with the software package Paravision (Bruker BioSpin Corp, Billerica, MA, USA), a 15-cm-diameter gradient coil with 450-mT/m maximum gradient strength (Resonance Research Corp, Billerica, MA, USA) and custom 10-channel phased-array receive coil which conformed to the 3D printed head holder. Radiofrequency transmission was accomplished with a 16-rung high-pass birdcage coil. Functional imaging was performed in a single session with 4-8 functional runs (at 512 volumes each) with the following parameters: TR = 2000 ms, TE = 22.2 ms, flip angle = 70.4, field of view = 28 × 36 mm, matrix size = 56 × 72, voxel size = 0.5 × 0.5 × 0.5 mm, slices = 38. Two sets of spin-echo EPI with an opposite phase-encoding direction (left-right and right-left) were collected for the EPI-distortion correction (TR = 3000 ms, TE = 0.44 ms, flip angle = 90 degrees, FOV = 28 × 36 mm, matrix size = 56 × 72, voxel size = 0.5 × 0.5 × 0.5 mm, axial slices = 38). T2-weighted structural image scans were acquired for each animal during one of the awake sessions with the following parameters: TR = 6000 ms, TE = 9 ms, flip angle = 90°, FOV = 28 × 36 mm, matrix size = 112 × 144, slices = 38, voxel size = 0.25 × 0.25 × 0.5 mm, number of averages = 8.

*Image preprocessing.* For squirrels, rats, and marmosets, data were similarly processed with custom preprocessing pipelines using the Analysis of Functional NeuroImages (AFNI)[44] and FMRIB Software Library (FSL)[45] software packages. Raw functional images were converted to NifTI format using dcm2niix[46] and reoriented from the sphinx position using FSL. The images were then despiked (AFNI's 3dDespike), and volume registered to the middle volume (AFNI's 3dvolreg). The motion parameters from volume registration were stored for later use with nuisance regression. For the rats, images were smoothed by a 1 mm full-width at half-maximum Gaussian kernel to reduce noise (AFNI's 3dmerge); for the larger squirrel and marmoset brains, a 1.5 mm kernel was used. An average functional image was then calculated for each run and registered (FSL's FLIRT) to each animal's T2-weighted image—the 4D time-series data was transformed using this matrix. T2-weighted images were manually skull-stripped (including the olfactory bulb in all three species), and this mask was applied to the functional images.

Each individual animal's T2-weighted images were non-linearly registered to high-resolution anatomical templates: for squirrels, images were registered to the higher-resolution anatomical image from squirrel 5 (voxel size = 0.2 × 0.2 × 0.2 mm). For rats, images were registered to the anatomical image provided in template space (voxel size = 0.05 × 0.05 × 0.05 mm)[47]. For marmosets, images were registered to the NIH marmoset brain atlas (voxel size = 0.2 × 0.2 × 0.2 mm)[48]. Functional images from all three species were bandpass filtered between 0.01 and 0.1 Hz.

*Fingerprinting.* For frontal regions, seed analyses were conducted between the mean time course within each seed region and every other voxel in the brain (with the nuisance regressors described above). Group functional connectivity maps (Z score maps) were then calculated for each of the frontal seeds in the three species. For rats, seeds (0.9 mm isotropic cubes) were placed in frontal areas Fr3, FrA, M2, LO, and MO based on the atlas (see Fig. 2, red squares). For squirrels, no atlas is available; therefore, 6 seeds (1.6 mm isotropic cubes) were placed in the frontopolar cortex (see Fig. 2, red squares) across area F based on a paper from the Kaas lab[26]. For marmosets, seeds (single voxels) were placed in frontal area 6 VA, 6DR, 8AD, 8AV, 45, 47 L, 46D, and 46 V (see Fig. 2, red squares) using our open-access resource (https://marmosetbrainconnectome.org)[41]—given the high statistical

power of this functional connectivity dataset, averaging across multiple voxels was not necessary.

We then specified eight common regions extrinsic to the frontal cortex in all three species (placement shown in Fig. 2 as blue squares). For each species, regions of interest were manually drawn in the posterior parietal cortex (PPC), posterior cingulate cortex (PCC), S1, M1, insula (Ins), striatum, superior colliculus (SC), and pulvinar (Pul). For the rats, these regions of interest were 0.9 mm isotropic cubes; in the squirrels and marmosets, 1.5 mm isotropic cubes. The placement of the cubes was based on definitions of these areas in the rat and marmoset atlas and the squirrel cortical parcellation by Wong and Kaas[23]. We also systematically translated the seed regions about the original anatomically defined centroid in each species and found that the placement had minimal impact on the overall fingerprint pattern (Supplementary Figs. 7, 8 and 9 for rats, squirrels, and marmosets respectively).

With the eight regions of interest defined in each respective species, we then extracted the mean connectivity values within these regions (with variance shown in Supplementary Fig. 10)—these values constituted the fingerprints, with a separate fingerprint for each seed region, for each species. To compare across the species, we normalized the fingerprint to a range between 0 (weakest connection with any of the target regions) and 1 (strongest connection with any of the target regions). We thus compare a connectivity pattern with target areas rather than absolute strength in any given species[23]. For the comparisons, we calculated the multidimensional cosine similarity across the matrix of functional connectivity fingerprints[23]—intuitively similar to a correlation value. The cosine similarity analysis provided an index of how similar or different the interareal fingerprint patterns were. By comparing the cosine of the angle between vectors (i.e., fingerprints), the cosine similarity metric indexes how similar the orientation of a set of vectors are in normalized space, with high similarity values indicating similar fingerprints (i.e., vectors in the same direction) and low scores indicating dissimilar fingerprints (i.e., vectors of diverging direction). By plotting the fingerprints in spider plots, we show the specific regions where the fingerprints differ. We applied this technique to compare squirrel, rat, and marmoset frontal FC patterns with the eight extrinsic regions.

**Statistics and reproducibility.** Permutation testing was used to test for statistical differences between fingerprints across the species. Permutation tests were performed via in-house code written in Matlab. Pair-wise comparisons of fingerprints were performed for each seed region of interest by first randomly dividing individual (i.e., scan-level) fingerprints into two groups (with fingerprints from each of two species randomly forming the comparison distributions), group-wise averaging, then normalization of the fingerprints to a range of 0 (weakest connection with any of the target regions) and 1 (strongest connection with any of the target regions). Cosine similarity was then calculated across species. This process was iterated 10,000 times, yielding a distribution of cosine similarity values that trended towards 1. A cosine similarity value smaller than the lowest 1 percentile of permuted cosine similarity values was defined as a significantly different fingerprint comparison.

**Reporting summary.** Further information on research design is available in the Nature Research Reporting Summary linked to this article.

## Data availability
Data available upon reasonable request from the authors.

## Code availability
Code available upon reasonable request from the authors.

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

## Acknowledgements

Support was provided by the Canadian Institutes of Health Research (FRN 148365) and the Canada First Research Excellence Fund to BrainsCAN. This work was supported in part by the Pennsylvania Department of Health Commonwealth Universal Research Enhancement (C.U.R.E.) Tobacco Appropriation Funds – Phase 18 (SAP 4100083102). We wish to thank Cheryl Vander Tuin and Hannah Pettypiece for animal preparation and care and Dr. Alex Li and Dr. Diego Szupak for scanning assistance.

## Author contributions

D.J.S., K.M.G., M.B., A.C.S., and S.E. designed and conducted research. D.J.S. and S.E. analysed data. D.J.S. and S.E. wrote the manuscript. D.J.S., K.M.G., M.B., A.C.S., and S.E. edited the manuscript.

## Competing interests

The authors declare no competing interests.
