## [Peer Review File · Communications Biology]

Reviewers' comments:

Reviewer #1 (Remarks to the Author):

-- Summary --

The present manuscripts compares connectivity profiles of frontal areas with parietal areas across squirrels, rats and macaque, with the aim of showing the similarity/dissimilarity of frontoparietal connectivity in mammalia. This is a very interesting and intriguing study which uses a somewhat understudied animal model with great potential for shedding light onto the the phenomenon of convergent evolution.

While I am in favor of seeing this manuscript published, I have some major issues with it that needs to be addressed by the authors prior to further consideration.

-- major comments --

1) In your results you appear to show significant differences (white stars) in a high number of pairs, yet your interpretation focuses on the fact that many pairs (between squirrel and macaque) are highly similar. This does not fare well with each other. Either they are similar or they are significantly different.

As I'll outline below, I do believe (hopefully) that your permutation test might not be set up correctly for testing interspecies differences. Please see my comment in the result section, as this might solve this issue and thus solve some of my comments in about your discussion as well.

-- general --

2) Please reconsider changing phrases/statments about "frontoparietal networks" to frontoparietal connectivity, as you do not investigate networks in the sense of data-driven network definitions. Just computing the connectivity from some more or less randomly selected seeds in frontal and parietal areas, does not make it a network - and given the lack of studies on brain parcellation or data driven network identification in squirrels, I'm uncertain if we even know at this point what areas in the squirrel brain would even be part of a frontoparietal network.

-- minor/detailed comments --

- abstract -

3) line 28: "we sought to determine if the species-specific lifestyles could explain the previously reported differences in frontoparietal network connectivity between rodents and primates."
-> after this sentence, I expected a comparison between different kinds of squirrels (some that are diurnal and some that are nocturnal). Please consider rephrasing! If "lifestyles" are your core motivation, the study design should better reflect this. Instead you could consider highlighting the squirrel model (with all it's benefits compared to rats AND it's benefits for letting us understand more about the phylogenetic tree and brain evolution (e.g. see <https://www.sciencedirect.com/science/article/pii/S1053811920311708>).

4) Line 34: "We also mapped the interareal connectivity patterns of these regions in each species, utilizing common cortical and subcortical brain areas across the three species."

-> To what extend "common" cortical/subcortical areas across species are actually homologous might be debatable... This sentence does not reflect that you used a fingerprinting technique! Please consider rephrasing to make this clearer. Furthermore, I'm uncertain why you say "we also mapped", when this is in fact the main analyses.

- Introduction -

5) Line 71: "Therefore, it is prudent to investigate the functional cortical organization in a rodent species that has a lifestyle more like that of many primate species."

-> I do generally agree with the logic of this statement, however, it makes me wonder what

results we got so far on the impact of fundamental lifestyle features on the functional neural architecture. I thus suggest adding some literature that investigated this relationship. (This also goes along with one of my comments above to maybe focus more on highlighting the benefits of studying squirrel brains)

-- results --

6) The results reported in Figure 2 feel very unintuitive, given that some of the pairs with high cosine similarity are reported as significantly different between species. Could you please provide the plots of the permuted cosine distribution (Maybe just for review purposes)? If I understood correctly, for example for fig2B: 6DR vs F4, such a plot should show a very strong rightward distribution with more than 99% of permuted values to the right of your observed cosine similarity.

Furthermore, I did not clearly understand what you permuted. If you want to test for differences between the species, it must have been a permutation of the label (marmoset/squirrel). Hence, I believe either a) you might have misinterpreted whether or not the permutation tests here actually tests for inter-species differences or b) I'm simply misunderstanding what exactly you permuted. In this sense: please explain what is meant with "individuals" in line 382. If you permuted over individuals WITHIN species and then calculating the cosine similarity between the species, the permutation test you used does not test for differences between species because: H_0 = observed cosine similarity is not better/worse than 99% of the distribution of cosine similarity calculated between species A and species B when individuals were shuffled. However, what you want (in order to test for differences between species) is: H_0 = observed cosine similarity (measure of similarity between group) is not better/worse than 99% of the distribution of cosine similarity calculated between two random groups formed from species A and species B.

I hope this comment helps to adjust the analyses for validly comparing the connectivity profiles.

-- discussion --

7) Line 141: "Our results demonstrate that grey squirrels, but not rats, possess a prominent frontoparietal network organization."

-> I'm uncertain what is meant with "prominent" in this context. You did not test IF there is a frontoparietal network organization, but only whether or not there are differences between the species in frontoparietal connectivity profiles. Please consider rephrasing.

8) Line 160: "This finding supports the notion that frontal areas in rats resemble primate premotor but not the prefrontal cortex"

-> please provide literature for this notion or rephrase to make clear that this is your interpretation.

9) Line 162: "The pattern, however, was entirely different for the comparison of connectivity profiles between squirrels and marmosets. We found that frontopolar regions in the squirrel showed good correspondence with lateral frontal cortex areas 6DR, 8aD, 8aV, and 47 in marmosets"

-> IF your permutation test truly tests for significant differences (which I'm not certain about, as outlined above), your current results do, in my opinion, not support this statement. "good correspondence" does not fare well with "significantly different", and pointing at the number of high cosine similarity pairs, does not solve this.

-- methods --

10) Line 214: subjects..

-> how many rats were included?

-> given the low number of subjects, it might make sense to also report the variance in connectivity pattern for each species, maybe as supplementary material. This will also be

beneficial for other researchers that are interested in the general strength of frontoparietal connectivity and might thus increase your citation rate.

11) Line 337: "Each individual animal's T2-weighted images were non-linearly registered to high-resolution anatomical templates [...]"

-> please report the resolution of the different reference spaces used for registration.

12) Line 344: "fingerprinting"

I'm uncertain about the wording here... are these "fingerprints" simply the functional connectivity matrices from your 8 areas, averaged across members of a species? Please elaborate.

13) line 362: "We also systematically translated these regions about the original anatomically defined centroid in each species and found that the placement had minimal impact on the overall fingerprint pattern."

-> This is quite interesting! Could you please add this analyses as supplementary material?

14) Line 366: "To compare across the species, we normalized the fingerprint to a range between 0 (weakest connection with any of the target regions) and 1 (strongest connection with any of the target regions).

-> again: what is meant with fingerprint in this case?

If this is the entire connectivity matrix, the values should already be between 0 and 1 as the elements in the matrix should represent the correlation coefficient (spearman/pearson)... this should also be true, if "fingerprint" is supposed to mean a column of the connectivity matrix (connectivity profile of one region with each other ROIs).

I believe my confusion comes from my uncertainty about how you calculated the mean connectivity values (what is the metric/statistic here?) and my lack of understanding what you label as "fingerprint".

Please elaborate.

Reviewer #2 (Remarks to the Author):

The study of Schaeffer et al. focused on frontoparietal network connectivity which is an important feature of primate cortical organization. Knowing that the previous studies were carried out in rats and mice which are nocturnal and terrestrial animals unlike most primates, the present work compared the frontoparietal connectivity between a primate, marmoset, and two species of rodents, rats and grey squirrels (diurnal and arboreal) using ultra-high field resting state fMRI. The results showed that grey squirrels, contrary to rats, possess a frontoparietal network similar to the connectivity pattern of marmoset. To conclude, the acquisition of an arboreal lifestyles in grey squirrels and marmosets, without having a common arboreal ancestor, has shown that the expansion of the visual system and the formation of a frontoparietal network architecture might reflect convergent evolution.

The paper is interesting, well written and overall pointing out a relationship between marmoset and grey squirrel on the organisation of frontoparietal network. To facilitate the understanding of readers, I do have a number of comments below which I think may help convey the core message more clearly.

1. In the abstract, the authors should standardize the name of the 3 species by putting the latin name.

2. To facilitate the understanding of readers who are not always specialists in the evolution of species, I think to add a phylogenetic tree in the manuscript could help to place the different species noted in the evolutionary context.

3. In the introduction part (lines 73-76), the authors showed similarities between squirrels and marmosets but, they did not indicate values (or ranges) for body sizes (body mass).

4. I think it would be relevant to make a summary in the form of a table for the similarities and

differences between these 3 species.

5. In the discussion part, I was disappointed that the notion of evolutionary convergence was not more developed. Indeed, this point is very interesting and innovative especially in imaging studies and comparisons between species. I will advise the authors to open the discussion on the evolutionary convergence.

6. In the Materials and Methods part (lines 246 and 248), the database for the marmoset data indicates that 31 awake animals were used in this study. However, contrary to rats and grey squirrels, the authors did not give the weight and age information for the marmosets.

7. In the Materials and Methods part, I noticed that two types of MRI were used for MRI data of marmosets. Can the authors explain why they integrated data that did not come from the same system as for the other species? The two types of MRI (9.4 T and 7 T) do not have the same characteristics. Were the data acquired used to obtain the same information, to create the same figure (average between the 2 methods)? I think the authors should be more clear.

8. In the different figures, the authors should add the meaning of FrA, LO, MO, Fr3, M2, F1, F2, F3, F4, F5, F6, 6VA, 6DR, 8AD, 8AV, 45, 47, 46D and 46V for all figures.

Reviewer #3 (Remarks to the Author):

Major comments:

This study is a well-designed and well-executed comparative analysis of functional brain organization that provides important insights into long-standing questions about the evolution of prefrontal cortex in mammals. Although evidence has been accumulating for quite some time about the lack of/poor homology of primate granular PFC with rodent PFC (specifically, laboratory animal species, which are part of the rodent subfamily Murinae). This study takes a novel and clever approach by comparing the functional connectivity of PFC in another rodent species from a different taxon (Sciuridae) which shows convergent evolution in terms of behavioral niche as primates. The findings not only show compelling evidence for the convergent evolution of organizational specializations to PFC for diurnal, arboreal mammals, but characterizes the specific fronto-parietal architecture that may be implicated in the evolution of this lifestyle in mammals. The dataset is of an impressive quality and the sample sizes are appropriate. This study opens up exciting new doors for comparative neuroscience research on not only PFC and fronto-parietal architecture, but visual specializations and subcortical-visual cortex networks as well. I don't have any major suggestions to make to improve this study, it is excellent as is.

Minor comments:

Line 51: Should be "Scandentia", not "Scadentia"

Line 215: I assume by "black variants" the authors mean "melanistic"; melanistic is the more commonly used term for black *S. carolinensis*

Methods, Line 244 (marmosets): Please include the ages/age range of the marmosets in question.

We would like to thank the editors and reviewers for their time and effort on our manuscript.

We are grateful for the refinements offered by the reviewers – these constructive comments have substantially improved our manuscript. Below, please find each critique listed with a detailed response. Corresponding changes in the manuscript are indicated in red font.

Response to Reviewer # 1:

Reviewer #1 (Remarks to the Author):

-- Summary --

The present manuscript compares connectivity profiles of frontal areas with parietal areas across squirrels, rats and macaque, with the aim of showing the similarity/dissimilarity of frontoparietal connectivity in mammalia. This is a very interesting and intriguing study which uses a somewhat understudied animal model with great potential for shedding light onto the the phenomenon of convergent evolution.

While I am in favor of seeing this manuscript published, I have some major issues with it that needs to be addressed by the authors prior to further consideration.

Reply: We appreciate your detailed suggestions, particularly regarding the statistical comparison of the connectivity topologies across the different species. As you will find in the responses below, we have improved on our method of statistically comparing the connectivity topologies across species and have made the other requested changes in turn.

-- major comments --

1) *In your results you appear to show significant differences (white stars) in a high number of pairs, yet your interpretation focuses on the fact that many pairs (between squirrel and macaque) are highly similar. This does not fare well with each other. Either they are similar or they are significantly different.*

As I'll outline below, I do believe (hopefully) that your permutation test might not be set up correctly for testing interspecies differences. Please see my comment in the result section, as this might solve this issue and thus solve some of my comments in about your discussion as well.

Reply: As detailed in response to your following comments, we have rephrased the results such that we refer to *significant differences*, rather than the extent of the *similarities*, *per se*. Further, we have clarified how our permutation testing works to test for interspecies differences and have corrected a missed step of normalization in this analysis that led to skewed permuted distributions (reply to comment # 6).

-- general --

2) *Please reconsider changing phrases/statments about "frontoparietal networks" to frontoparietal connectivity, as you do not investigate networks in the sense of data-driven network definitions. Just computing the connectivity from some more or less randomly selected seeds in frontal and parietal areas, does not make it a network - and given the lack of studies on brain parcellation or data driven network identification in squirrels, I'm uncertain if we even*

know at this point what areas in the squirrel brain would even be part of a frontoparietal network.

Reply: As requested, we have changed “frontoparietal networks” to “frontoparietal connectivity” throughout the manuscript (including the title).

-- minor/detailed comments --

- abstract -

3) line 28: "we sought to determine if the species-specific lifestyles could explain the previously reported differences in frontoparietal network connectivity between rodents and primates."

-> after this sentence, I expected a comparison between different kinds of squirrels (some that are diurnal and some that are nocturnal). Please consider rephrasing! If "lifestyles" are your core motivation, the study design should better reflect this. Instead you could consider highlighting the squirrel model (with all it's benefits compared to rats AND it's benefits for letting us understand more about the phylogenetic tree and brain evolution (e.g. see <https://www.sciencedirect.com/science/article/pii/S1053811920311708>).

Reply: Good point. We have removed this sentence in the abstract, as well as other statements referring to “lifestyles”. As per Reviewer # 2’s comment # 4, we have also included a phylogenetic tree (now Figure 1) and table (now Table 1) to highlight the similarities between primates and squirrels. Finally, as suggested, we have integrated the following text in the introduction to highlight the benefits of using squirrels for studying brain evolution.

Figure 1. Phylogenetic tree showing the timelines of divergence (MYA: million years ago) and lineage of rats (*Rattus norvegicus*), squirrels (*Sciurus carolinensis*) and marmosets (*Callithrix jacchus*).

Table 1. Comparison of relevant features in each model species (similarities in **bold**)

	Rat (Rattus norvegicus)	Grey squirrel (Sciurus carolinensis)	Common marmoset (Callithrix jacchus)
Order	Rodentia	Rodentia	Primates
Sleep/wake cycle	Nocturnal	Diurnal	Diurnal
Navigation behavior	Whisking and olfaction	Vision	Vision
Environment	Terrestrial	Arboreal	Arboreal
Body mass	250 – 500 g	300 – 710 g	250 – 650 g
Brain mass	1.8 g	7.6 g	7.8 g

Lines 56-61: “Here, we set out to better understand how frontoparietal functional connectivity in a mammal outside of the primate order – the Eastern Gray Squirrel (grand order Glires) – compares both primates (grand order Euarchonta) and rodents (grand order Glires). Specifically, by comparing patterns of connectivity in an animal that has evolved in a similar environmental niche to primates, we sought to gain insight on how their brain adapted to within the constraints of its evolutionary history.”

4) Line 34: *"We also mapped the interareal connectivity patterns of these regions in each species, utilizing common cortical and subcortical brain areas across the three species."*

-> *To what extent "common" cortical/subcortical areas across species are actually homologous might be debatable... This sentence does not reflect that you used a fingerprinting technique!*

Please consider rephrasing to make this clearer. Furthermore, I'm uncertain why you say "we also mapped", when this is in fact the main analyses.

Reply: Yes, we have changed this sentence to remove the implication of areal homology across species and have made it clear that comparisons of connectivity from the frontal seeds were made via fingerprinting analysis. Please see corrected sentence as follows:

Lines 32-34: “We utilized a “fingerprinting” analysis to compare interareal patterns of functional connectivity from seeds across frontal cortex in all three species.”

- *Introduction* -

5) *Line 71: "Therefore, it is prudent to investigate the functional cortical organization in a rodent species that has a lifestyle more like that of many primate species."*

-> I do generally agree with the logic of this statement, however, it makes me wonder what results we got so far on the impact of fundamental lifestyle features on the functional neural architecture. I thus suggest adding some literature that investigated this relationship. (This also goes along with one of my comments above to maybe focus more on highlighting the benefits of studying squirrel brains)

Reply: As stated in the reply to comment # 3 above we have added text to clarify that we are not targeting a comparison of lifestyle, but rather how brains adapt within the constraints of evolutionary history. We have also integrated accompanying literature.

-- *results* --

6) The results reported in Figure 2 feel very unintuitive, given that some of the pairs with high cosine similarity are reported as significantly different between species. Could you please provide the plots of the permuted cosine distribution (Maybe just for review purposes)? If I understood correctly, for example for fig2B: 6DR vs F4, such a plot should show a very strong rightward distribution with more than 99% of permuted values to the right of your observed cosine similarity.

Furthermore, I did not clearly understand what you permuted. If you want to test for differences between the species, it must have been a permutation of the label (marmoset/squirrel).

Hence, I believe either a) you might have misinterpreted whether or not the permutation tests here actually tests for inter-species differences or b) I'm simply misunderstanding what exactly you permuted. In this sense: please explain what is meant with "individuals" in line 382. If you permuted over individuals WITHIN species and then calculating the cosine similarity between the species, the permutation test you used does not test for differences between species because: H_0 = observed cosine similarity is not better/worse than 99% of the distribution of cosine similarity calculated between species A and species B when individuals were shuffled.

However, what you want (in order to test for differences between species) is:

H_0 = observed cosine similarity (measure of similarity between group) is not better/worse than 99% of the distribution of cosine similarity calculated between two random groups formed from species A and species B.

I hope this comment helps to adjust the analyses for validly comparing the connectivity profiles.

Reply: Thank you, we very much appreciate your detailed examination of our analysis. Indeed, we have modified (corrected, really) the permutation analysis. Explicitly, we identified an

oversight in the analysis of the original submission – that we did not normalize the functional connectivity values after each permutation before computing the cosine similarity values – this led to very skewed permuted distributions. This explains why the originally submitted Figure 2 was “very unintuitive,” with some high cosine similarity values (yellow cells) being significantly different alongside much lower cosine similar (green or blue cells) that did not meet the threshold for being statistically different. Please find the requested distributions below (e.g., marmoset 6DR vs squirrel F4 seeds, and also marmoset 8D with squirrel F2), before and after this change, as well as the corrected figure.

Original Submission

Corrected Analysis

Marmoset 6DR with Squirrel F4

Marmoset 6DR with Squirrel F4

Marmoset 8D with Squirrel F2

Marmoset 8D with Squirrel F2

Regarding what we are permuting, it is between two random groups formed from species A and species B. Please find a schematic below to clarify the analysis, along with the text added to the methods section as follows:

Lines 500 – 406: “Permutation testing was used to test for statistical differences between fingerprints across the species. Permutation tests were performed via in-house code written in Matlab. Pair-wise comparisons of fingerprints were performed for each seed region of interest by first randomly dividing individual (i.e., scan-level) fingerprints into two groups (with fingerprints from each of two species randomly forming the comparison distributions), group-wise averaging, then normalization of the fingerprints to a range of 0 (weakest connection with any of the target regions) and 1 (strongest connection with any of the target regions).”

-- discussion --

7) Line 141: "Our results demonstrate that grey squirrels, but not rats, possess a prominent frontoparietal network organization."

-> I'm uncertain what is meant with "prominent" in this context. You did not test IF there is a frontoparietal network organization, but only whether or not there are differences between the species in frontoparietal connectivity profiles. Please consider rephrasing.

Reply: We agree that this may not be the best word choice and thus have removed “prominent” throughout the manuscript as it refers to the frontoparietal connectivity in squirrels.

Lines 148-149: “Our results demonstrate that grey squirrels have frontoparietal connectivity topologies that better overlap with that of primates than to that of rats.”

8) Line 160: *"This finding supports the notion that frontal areas in rats resemble primate premotor but not the prefrontal cortex"*

-> *please provide literature for this notion or rephrase to make clear that this is your interpretation.*

Reply: We have clarified this statement, specifically referring to the first column in Figure 3A, as well as adding supporting literature. Please find modified text as follows:

Lines 164 – 166: “This finding supports the notion that frontal areas in rats resemble primate premotor but not the prefrontal cortex (Figure 3A; also see 13).”

9) Line 162: *"The pattern, however, was entirely different for the comparison of connectivity profiles between squirrels and marmosets. We found that frontopolar regions in the squirrel showed good correspondence with lateral frontal cortex areas 6DR, 8aD, 8aV, and 47 in marmosets"*

-> *IF your permutation test truly tests for significant differences (which I'm not certain about, as outlined above), your current results do, in my opinion, not support this statement. "good correspondence" does not fare well with "significantly different", and pointing at the number of*

high cosine similarity pairs, does not solve this.

Reply: You are correct – we tested for significant differences between fingerprints, rather than for similarities, *per se*. We have corrected this statement for accuracy:

Lines 167-169: “The pattern, however, was entirely different for the comparison of connectivity profiles between squirrels and marmosets. We found that frontopolar regions in the squirrel showed high cosine similarities values that, unlike rats, did not significantly differ from patterns of lateral frontal cortex areas 6DR, 8aD, 8aV, and 47 in marmosets.”

-- methods --

10) Line 214: *subjects..*

-> how many rats were included?

-> given the low number of subjects, it might make sense to also report the variance in connectivity pattern for each species, maybe as supplementary material. This will also be beneficial for other researchers that are interested in the general strength of frontoparietal connectivity and might thus increase your citation rate.

Reply: Data were collected from five rats. As requested, we have also plotted variance for each of the species (pasted below) and it is now included in the manuscript as Supplementary Figure 10. From this analysis, we can see the variance within a fingerprint is relatively homogenous, especially in rats and squirrels. Further, we can see the variance from marmosets was higher,

albeit the marmoset data was acquired awake and had a much higher n . The inhomogeneity of the marmoset variance tracks well with signal to noise ratio distributions from the receive coil, with higher SNR toward cortex that decreases toward the subcortical regions.

Lines 239-240: “Data were collected from five adult male Wistar rats aged 8-12 weeks and weighing from 250-350 g.”

Figure S10. Variance plotted for each fingerprint, in each species.

11) Line 337: "Each individual animal's T2-weighted images were non-linearly registered to high-resolution anatomical templates [...]"

-> please report the resolution of the different reference spaces used for registration.

Reply: As requested, we have integrated the resolution(s) of the template spaces into the methods section as follows:

Lines 354 – 359: “Each individual animal’s T2-weighted images were non-linearly registered to high-resolution anatomical templates: for squirrels, images were registered to the higher-resolution anatomical image from squirrel 5 (voxel size = 0.2 x 0.2 x 0.2 mm). For rats, images were registered to the anatomical image provided in template space (voxel size = 0.05 x 0.05 x 0.05 mm) (46). For marmosets, images were registered to the NIH marmoset brain atlas (voxel size = 0.2 x 0.2 x 0.2 mm) (47).”

12) Line 344: *"fingerprinting"*

I'm uncertain about the wording here... are these "fingerprints" simply the functional connectivity matrices from your 8 areas, averaged across members of a species? Please elaborate.

Reply: We have clarified the meaning of “fingerprint” in the text as follows:

Lines 387 – 389: “With the eight regions of interest defined in each respective species, we then extracted the mean connectivity values within these regions – these values constituted the fingerprints, with a separate fingerprint for each seed region, for each species.”

13) line 362: *"We also systematically translated these regions about the original anatomically defined centroid in each species and found that the placement had minimal impact on the overall fingerprint pattern."*

-> This is quite interesting! Could you please add this analyses as supplementary material?

Reply: Yes, we have added these comparisons as Supplementary Figures 7, 8 and 9 (pasted below). Explicitly, for each species, we translated each of the frontal seed regions by one (functional) voxel in all three dimensions (+/-). O = original; L = left; R = right; A = anterior; P = posterior; D = dorsal; V = ventral. Note that for the regions with the greatest variability (squirrel F3, in particular), moving the seed by one voxel may have moved the seed partially out of parenchyma.

Supplementary Figure 7 (rats)

Supplementary Figure 8 (squirrels)

Supplementary Figure 9 (marmosets)

Lines 380-383: “We also systematically translated the seed regions about the original anatomically defined centroid in each species and found that the placement had minimal impact on the overall fingerprint pattern (Figures S7, S8, and S9 for rats, squirrels, and marmosets respectively).”

14) Line 366: "To compare across the species, we normalized the fingerprint to a range between 0 (weakest connection with any of the target regions) and 1 (strongest connection with any of the target regions).

-> again: what is meant with fingerprint in this case?

If this is the entire connectivity matrix, the values should already be between 0 and 1 as the elements in the matrix should represent the correlation coefficient (spearman/pearson)... this should also be true, if "fingerprint" is supposed to mean a column of the connectivity matrix (connectivity profile of one region with each other ROIs).

I believe my confusion comes from my uncertainty about how you calculated the mean connectivity values (what is the metric/statistic here?) and my lack of understanding what you label as "fingerprint".

Please elaborate.

Reply: As detailed in our response above (comment # 12), we have clarified what constitutes a fingerprint in the manuscript. Likely what is contributing to the confusion here is that the *lowest* value in the fingerprint (which is a z value) becomes 0 and the *highest* value becomes 1 (notice

all fingerprints report exactly one value at both zero, and another at one). In this way, we are comparing patterns of connectivity, rather than Pearson correlation values (for example).

Lines 386 -390: “To compare across the species, we normalized the fingerprint to a range between 0 (weakest connection with any of the target regions) and 1 (strongest connection with any of the target regions). We thus compare a connectivity pattern with target areas rather than absolute strength in any given species (26).”

Response to Reviewer # 2:

Reviewer #2 (Remarks to the Author):

The study of Schaeffer et al. focused on frontoparietal network connectivity which is an important feature of primate cortical organization. Knowing that the previous studies were carried out in rats and mice which are nocturnal and terrestrial animals unlike most primates, the present work compared the frontoparietal connectivity between a primate, marmoset, and two species of rodents, rats and grey squirrels (diurnal and arboreal) using ultra-high field resting state fMRI. The results showed that grey squirrels, contrary to rats, possess a frontoparietal network similar to the connectivity pattern of marmoset. To conclude, the acquisition of an arboreal lifestyles in grey squirrels and marmosets, without having a common arboreal ancestor, has shown that the expansion of the visual system and the formation of a frontoparietal network architecture might reflect convergent evolution.

The paper is interesting, well written and overall pointing out a relationship between marmoset and grey squirrel on the organisation of frontoparietal network. To facilitate the understanding of readers, I do have a number of comments below which I think may help convey the core message more clearly.

Reply: Thank you for these clarifications. As you will find below, we have made your suggested changes in turn:

1. In the abstract, the authors should standardize the name of the 3 species by putting the latin name.

Reply: As requested, we have added the Latin name for each species into the abstract:

Lines 32-34: “We used ultra-high field resting-state fMRI data to compute and compare the functional connectivity (FC) patterns of frontal regions in grey squirrels (*Sciurus carolinensis*), rats (*Rattus norvegicus*), and marmosets (*Callithrix jacchus*).”

2. To facilitate the understanding of readers who are not always specialists in the evolution of species, I think to add a phylogenetic tree in the manuscript could help to place the different species noted in the evolutionary context.

Reply: A phylogenetic tree (now Figure 1) has been added to the introduction section. Please also find the figure pasted below:

Figure 1. Phylogenetic tree showing the timelines of divergence (MYA: million years ago) and lineage of rats (*Rattus norvegicus*), squirrels (*Sciurus carolinensis*) and marmosets (*Callithrix jacchus*).

3. In the introduction part (lines 73-76), the authors showed similarities between squirrels and marmosets but, they did not indicate values (or ranges) for body sizes (body mass).

Reply: As per the next comment (comment no. 4), we have now included a table showing similarities across the species, including ranges of body mass.

4. I think it would be relevant to make a summary in the form of a table for the similarities and differences between these 3 species.

Reply: Thank you for this suggestion; Table 1 has been added to the introduction (pasted below):

Line 76-78: “Therefore, it is prudent to investigate the functional cortical organization in a rodent species that has a lifestyle more like that of many primate species (See Table 1 for comparison).”

Table 1. Comparison of relevant features in each model species (similarities in **bold**)

	Rat (Rattus norvegicus)	Grey squirrel (Sciurus carolinensis)	Common marmoset (Callithrix jacchus)
Order	Rodentia	Rodentia	Primates
Sleep/wake cycle	Nocturnal	Diurnal	Diurnal
Navigation behavior	Whisking and olfaction	Vision	Vision
Environment	Terrestrial	Arboreal	Arboreal
Body mass	250 – 500 g	300 – 710 g	250 – 650 g
Brain mass	1.8 g	7.6 g	7.8 g

5. In the discussion part, I was disappointed that the notion of evolutionary convergence was not more developed. Indeed, this point is very interesting and innovative especially in imaging

studies and comparisons between species. I will advise the authors to open the discussion on the evolutionary convergence.

Reply: We agree that this is an important point to get across. Please find the added text, which we end the Discussion section with.

Lines 200-225: “Grey squirrels and marmosets have arrived at similar semiarboreal lifestyles. However, they show no common arboreal ancestor, indicating that the expansion of the visual system and the formation of frontoparietal connectivity architecture might have evolved along similar lines driven by comparable ecological niches as an example of convergent evolution (30). A diurnal activity pattern and an arboreal habitat may have favored color vision, high visual acuity, and depth perception for navigation in both species. At present, it cannot be ruled out that a frontoparietal network organization is typical in many rodent species and that the subfamily Muridae, which includes Old World rats and mice, lost this organization due to their adaptation to a terrestrial and mainly nocturnal lifestyle. It has been recently posited that evolution of the dorsal and ventral visual processing streams in both primates and squirrels was the result of occupying new, diurnal environments after the extinction of dinosaurs.

Rats and mice are powerful animal models for neuroscience. They are easy to breed, and powerful genetic tools have been developed that allow targeted probing and dissection of circuit functions in mice. However, our findings here do not support the idea that rats have an organization of frontoparietal connectivity similar to primates. Based on the results presented here – that strong frontoparietal connectivity is present in *Sciurus carolinensis*, but is lacking in their Muridae counterparts, *Rattus norvegicus* (at least to the extent that it overlaps with primate

connectivity patterns) – we highlight the importance of environmental niche in neuroscientific modelling species despite the constraints of evolutionary history. Indeed, evolutionary proximity is critically important, but evolutionary convergence may also be leveraged, perhaps improving the potential translatability of results yielded from preclinical modelling species. This reigns especially true for inquiries in visual neuroscience. Indeed, even though squirrels and primates have been evolving independently for nearly 100 million years (Figure 1) from ancestral insectivores (CITE, LANE 1971), we found that grey squirrels show a frontoparietal architecture with comparable patterns to the marmoset brain, opening the possibility of using squirrels as a rodent model for vision and comparative explorations of frontoparietal network organization and function.”

6. In the Materials and Methods part (lines 246 and 248), the database for the marmoset data indicates that 31 awake animals were used in this study. However, contrary to rats and grey squirrels, the authors did not give the weight and age information for the marmosets.

Reply: Age and weight ranges have been added as follows:

Line 247-248: “This database contains resting-state fMRI data from 31 awake marmosets (Callithrix jacchus, 8 females; age: 14 -115 months; weight: 240 - 625 g) that were acquired at the University of Western Ontario (5 animals) and the National Institutes of Health (26 animals).”

7. In the Materials and Methods part, I noticed that two types of MRI were used for MRI data of marmosets. Can the authors explain why they integrated data that did not come from the same system as for the other species? The two types of MRI (9.4 T and 7 T) do not have the same characteristics. Were the data acquired used to obtain the same information, to create the same figure (average between the 2 methods)? I think the authors should be more clear.

Reply: The rats, squirrels, and 5 of the 31 marmosets were acquired on the same 9.4 T MRI at the University of Western Ontario. The decision was made to combine the 9.4 T and 7 T marmoset data (i.e., awake RS-fMRI data from the University of Western Ontario with that from the 7 T MRI at the National Institutes of Health) to increase statistical power and heterogeneity of the population. Despite the differences in field strength, head fixation type, different coils (transmit, receive, and gradient coils), and slightly different acquisition parameters, the datasets are indeed comparable, as detailed in our recent manuscript (<https://www.sciencedirect.com/science/article/pii/S1053811922001598?via%3Di>). In particular, the registrations, functional connectivity topologies, and power were comparable across the datasets. Please see the supporting figures below from Schaeffer et al., 2022, *NeuroImage*, where sub-01 - 05 were acquired at 9.4T and sub-06 - 32 were acquired at 7T. Note Figure 6 pasted below is particularly relevant, showing that randomly selected runs from either the 9.4T or 7T show very similar statistical power (i.e., that to produce a reliable FC map).

Individual animal BOLD registrations to template

Figure 2. (from Schaeffer et al., 2022). Individual marmoset's registration to the Marmoset Brain Mapping (MBM) template space. For each run, mean BOLD images are computed for each animal, then linearly registered to the individual's T2-weighted structural anatomy. Next, these linearly registered images are nonlinearly registered to the template based on the registration of each animals' T2-weighted image to the T2-weighted template space. The first image shows the MBM template image, with the gray matter boundaries outlined in red. The following images show this outline overlaid on each animals' final registration result.

Fig. 3. (from Schaeffer et al., 2022). FC topologies (from an 8aV seed in the right hemisphere) for all marmosets currently contributing to the resource. The first image shows the group topology without a statistical (lower) threshold applied. The following images show FC from each animal from the same 8aV seed.

Fig. 4. (from Schaeffer et al., 2022). FC topologies (with the mean time course of area 8aD as a seed) demonstrate the reproducibility of the default mode network at the subject level. The first

image shows the group topology without a statistical (lower) threshold applied. The following images show FC from each animal from the same 8aD seed. The white outlines on each image show the regional hubs (i.e., areas of strong FC) of the default mode network, drawn from the group-level map. ACC = anterior cingulate cortex; PCC = posterior cingulate cortex; PPC = posterior parietal cortex.

Fig. 6. (from Schaeffer et al., 2022). (A) shows the power of the correlation maps (using the average time course from gray matter voxels as a seed) as a function of randomly selecting a given number of runs. The bottom row shows the power (variance) for both the NIH data (7 T) and UWO data (9.4 T). (B) shows the effect of the number of randomly selected runs on the resultant correlation topology.

8. In the different figures, the authors should add the meaning of FrA, LO, MO, Fr3, M2, F1, F2, F3, F4, F5, F6, 6VA, 6DR, 8AD, 8AV, 45, 47, 46D and 46V for all figures.

Reply: Done for all figures.

Response to Reviewer # 3:

Reviewer #3 (Remarks to the Author):

Major comments:

This study is a well-designed and well-executed comparative analysis of functional brain organization that provides important insights into long-standing questions about the evolution of prefrontal cortex in mammals. Although evidence has been accumulating for quite some time about the lack of/poor homology of primate granular PFC with rodent PFC (specifically, laboratory animal species, which are part of the rodent subfamily Murinae). This study takes a novel and clever approach by comparing the functional connectivity of PFC in another rodent species from a different taxon (Sciuridae) which shows convergent evolution in terms of behavioral niche as primates. The findings not only show compelling evidence for the convergent evolution of organizational specializations to PFC for diurnal, arboreal mammals, but characterizes the specific fronto-parietal architecture that may be implicated in the evolution of this lifestyle in mammals. The dataset is of an impressive quality and the sample sizes are appropriate. This study opens up exciting new doors for comparative neuroscience research on not only PFC and fronto-parietal architecture, but visual specializations and subcortical-visual cortex networks as well. I don't have any major suggestions to make to improve this study, it is excellent as is.

Reply: Thank you, we appreciate your positive summary.

Minor comments:

Line 51: Should be “Scandentia”, not “Scadentia”

Reply: Thank you for catching this error – we have corrected it throughout the manuscript.

Line 215: I assume by “black variants” the authors mean “melanistic”; melanistic is the more commonly used term for black S. carolinensis

Reply: Yes, we’ve corrected “black” with “melanistic”.

Methods, Line 244 (marmosets): Please include the ages/age range of the marmosets in question.

Reply: Age and weight ranges have been added as follows:

Line 247-248: “This database contains resting-state fMRI data from 31 awake marmosets (Callithrix jacchus, 8 females; age: 14 -115 months; weight: 240 - 625 g) that were acquired at the University of Western Ontario (5 animals) and the National Institutes of Health (26 animals).”

REVIEWERS' COMMENTS:

Reviewer #1 (Remarks to the Author):

My sincere apologies for the delay!

Thank you for the productive and extensive revision.

All of my comments have been addressed satisfyingly. After reading your the current version of the manuscript, I got no further comments. Therefore, I'm very much looking forward to seeing a published version of your study!

Reviewer #2 (Remarks to the Author):

I would like to thank and congratulate the authors for their work on this new version of their manuscript. The authors have responded to all my comments and made the necessary additions for a better understanding. I have no further comments, the article is publishable.